# Treatment of Sjögren’s Syndrome with Mesenchymal Stem Cells: A Systematic Review

**DOI:** 10.3390/ijms221910474

**Published:** 2021-09-28

**Authors:** Najwa Chihaby, Marie Orliaguet, Laëtitia Le Pottier, Jacques-Olivier Pers, Sylvie Boisramé

**Affiliations:** 1UFR d’Odontologie, University of Western Brittany, 29200 Brest, France; najwa.chihaby@laposte.net (N.C.); Marie.Orliaguet@etudiant.univ-brest.fr (M.O.); laetitia.lepottier@univ-brest.fr (L.L.P.); sylvie.boisrame@chu-brest.fr (S.B.); 2CHU de Brest, 29609 Brest, France; 3Inserm, LBAI, University of Western Brittany, UMR1227, 29609 Brest, France

**Keywords:** mesenchymal stem cells, Sjögren’s syndrome, oral sicca, xerostomia

## Abstract

Mesenchymal stem cells (MSCs) are ubiquitous in the human body. Mesenchymal stem cells were initially isolated from bone marrow and later from other organs such as fatty tissues, umbilical cords, and gingiva. Their secretory capacities give them interesting immunomodulatory properties in cell therapy. Some studies have explored the use of MSCs to treat Sjögren’s syndrome (SS), a chronic inflammatory autoimmune disease that mainly affects exocrine glands, including salivary and lacrimal glands, although current treatments are only palliative. This systematic review summarizes the current data about the application of MSCs in SS. Reports show improvements in salivary secretions and a decrease in lymphocytic infiltration in salivary glands in patients and mice with SS after intravenous or infra-peritoneal injections of MSCs. MSC injections led to a decrease in inflammatory cytokines and an increase in anti-inflammatory cytokines. However, the intrinsic mechanism of action of these MSCs currently remains unknown.

## 1. Introduction

Sjögren’s syndrome (SS), also called sicca syndrome, is a chronic inflammatory autoimmune disease that mainly affects women. It was first described in 1926 by Henri Gougerot, a French dermatologist, followed by Henrik Sjögren, a Swedish ophthalmologist, in 1933. This disease affects the functions of exocrine glands, causing dry mouth (hyposialia or even asialia) and dry eye (xerophthalmia). SS-induced hyposialia increases the risk of tooth decay, periodontal disease, and fungal infections [1]. Other extra-glandular organs such as skin can also be affected, and the patient’s quality of life can be degraded. SS can be observed as primary SS or secondary to other autoimmune diseases such as rheumatoid arthritis, systemic sclerosis, and polymyositis [2,3].

Immunologically, SS results from the infiltration of T lymphocytes into exocrine glands followed by a predominance of B lymphocytes in the advanced stages of the disease [4]. These T cells produce cytokines directed toward a cellular reaction as T helper 1 (Th1) and secondly toward a humoral response (Th2). Th1 and Th17 cells thus initiate the pathogenesis of SS, and as the disease progresses, Th2 and follicular helper T (Tfh) cells predominate [5]. B cells also play a role in cytokine and autoantibody production as well as antigen presentation. Regulatory B cells induce an increase in regulatory T (Treg) cells and a decrease in pro-inflammatory cytokines [4].

To date, there is no etiological treatment for this pathology, and currently, the management of SS is based primarily on the management of symptoms, e.g., the prescription of pilocarpine hydrochloride, a muscarinic receptor agonist, to improve salivary secretion [6].

In this context, the use of mesenchymal stem cells (MSCs) as a therapeutic approach to treating SS has been assessed and was previously reviewed [7,8,9]. MSCs are undifferentiated cells that can give rise to one or more cell lines while ensuring the renewal of their own population. They are characterized by their capacity for self-renewal and differentiation, but they also play an important role in immunomodulation. The pro-angiogenic, anti-inflammatory, and immunomodulatory properties of MSCs make them ideal candidates for targeted cell therapy aimed to restore immune functions [10]. Indeed, MSCs exert an immunomodulatory action on T and B cells, dendritic cells, and natural killers. Several studies in patients with autoimmune diseases such as rheumatoid arthritis or systemic lupus erythematosus have shown promising results [11].

Thus, the aim of this systematic review is to analyze current scientific knowledge on the application of mesenchymal stem cells in sicca manifestations such as SS.

## 2. Materials and Methods

### 2.1. Research Strategy

A comprehensive analysis of the literature was performed using databases such as PubMed, Web of Science, and Cochrane to inform this literature review.

The Mesh keywords used were “MSCs and oral sicca”, “MSCs and xerostomia”, and “MSCs and SS”.

### 2.2. Selection of Articles

Duplicate and irrelevant articles were eliminated. Studies were selected according to the following inclusion criterion: studies reporting the use of mesenchymal stem cells in sicca syndrome. The selection was refined according to the following exclusion criteria: studies concerning tissue engineering, other autoimmune diseases, damage to the salivary glands by radiotherapy, and ocular damage. After reading the title and abstract, articles were marked in a table with “included”, “excluded”, or “?”. No restrictions on the origin of the study or the year of publication interfered in the article selection.

### 2.3. Information Extraction

For each included study, the full text was reviewed to extract all the information, namely, origin of the MSCs, recipient, dose, follow-up, and results. Particular attention was paid to variations in salivary flow as well as cellular and molecular modifications.

## 3. Results and Discussion

### 3.1. Search Results

Research on Cochrane provided five clinical trials. As they did not meet the inclusion criteria, none of them were included in the analysis. In total, 247 articles were vetted, of which 128 were found on PubMed and 119 were found on Web of Science. After analyzing titles and results and eliminating irrelevant articles, 26 articles were selected (Figure 1).

#### Study Characteristics

Among the selected articles, different sources of MSCs were reported:Umbilical MSCs, umbilical cord MSCs (UMSCs, UCMSCs);Bone MSCs, bone marrow MSCs (BMSCs, BMMSCs);Dental pulp stem cells (DPSCs);Stem cells from exfoliated deciduous teeth (SHED);Murine embryonic MSCs (ME-MSCs);Olfactory ecto-MSCs, olfactory ecto-MSC-derived exosomes (OE-MSCs, OE-MSCs-Exos);Induced pluripotent stem cells (iPSC-MSCs);Salivary gland MSCs (SGMSCs).

Two articles focused on the function of bone morphogenic protein (BMP-6) in SGMSCs [12] and in BMMSCs [13] in a mouse model. The method of MSC transplantation in mice and humans appeared to be broadly consistent across the studies. Intravenous injections of MSCs were used in all studies [14,15,16,17,18,19,20,21,22,23,24,25,26,27,28,29], with the exception of one [30] in which MSCs were administered intraperitoneally. A variety of cocultures were used such as UCMSCs cocultured with peripheral blood mononuclear cells (PBMCs) [31,32], and MSCs cocultured with monocyte-derived dendritic cells [26] or CD4^+^ T cells and MSC coculture experiments [33].

Different mouse models were used in these studies. Non-obese diabetic mice (NOD/ShiLtJ), experimental Sjögren’s syndrome (ESS) mice, Murphy Roths Large (MRL/Lpr), and C57BL/6 mice were used as SS animal models; BALB/c mice and C57BL/6 mice served as the control or donor. All the in vivo MSC treatments in mouse models are summarized in Table 1.

All the allogeneic studies regarding the mouse models used stem cells from mice not affected by the disease. Xenografts were also found in mice insofar as they were human umbilical cord blood cells injected by veins or intraperitoneally. In human studies, an intravenous injection of allogeneic umbilical cord stem cells was performed. All the in vivo and in vitro studies in SS patients are summarized in Table 2.

### 3.2. Cellular and Molecular Changes

Cellular and molecular modifications induced by MSC administration in patients with SS and in mouse model of SS are shown in Figure 2.

#### 3.2.1. Role of Bone Morphogenic Proteins

Studies about the role of BMP6 on the immunomodulatory functions of MSCs showed an absence of correlation between pro-inflammatory cytokines or the presence of autoantibodies and the overexpression of BMP6. Anti-BMP6 targeted therapy increased the production of prostaglandin E2 (PGE2), but had no effect on other soluble factors (indoleamine 2,3-dioxygenase [IDO], nitric oxide [NO], transforming growth factor [TGF]-β). As Figure 3 shows, anti-BMP6 treatment induced an increase in PGE2 and a decrease in interferon (IFN) gamma and interleukin (IL)-17 via DNA-binding protein inhibitor (Id)1 [12,13]. BMP6 altered the immunomodulatory properties of SGMSCs by inhibition of proteins involved in Id1 [13].

Conversely, the amount of BMP4 was reduced in the BMMSCs of NOD/ShiLtJ mice compared to control mice and was likewise in patients with SS compared to healthy controls [18]. BMP4 was regulated by Id3, preventing the binding of E2A to the BMP4 promoter. Id3 controlled the immunosuppressive function of BMMSCs by inhibiting BMP4 expression, which induced decreased production of PGE2. Id3^−/−^ BMMSCs had a stronger immunoregulatory role than wild type BMMSCs. In this way, the Id3^−/−^ BMMSC treatment suppressed symptoms of SS more effectively than the wild type BMMSC treatment (Figure 3) [18].

In a coculture of MSCs and CD4^+^ T cells, an increased expression of PGE2, IL-6, and IDO was observed [16]. PGE2 mediated inflammation, with an immunosuppressive role in the activation and proliferation of T cells. PGE2 also inhibited the production of IFN gamma and IL-2 and conversely led to an increase in IL-5 [12].

#### 3.2.2. Impact on Pro- and Anti-Inflammatory Cytokines

Treatment with DPSCs caused a decrease in the expression of mRNA of pro-inflammatory cytokines (IL-2, IL-4, IL-6, IL-17a, IFN gamma) and an increase in anti-inflammatory cytokines IL-10 and TGF-β [22]. Likewise, there was a decrease in pro-inflammatory cytokines (IL-6, tumor necrosis factor [TNF]-α and IFN gamma) and a greater production of IL10, an anti-inflammatory cytokine, after coculture of UCMSCs and spleen mononuclear cells in NOD mice [30]. Therapy with BM-MSCs altered the concentration of cytokines and growth factors in NOD mice; thus, there was a decrease in IL-2 and IFN gamma and an increase in IL-6, hepatocyte growth factor (HGF)-β, IL-10, PGE2, and TGF-β [24]. After injection of stem cells from SHEDs in mice at different weeks (7, 14, and 21), a decrease in peripheral and local inflammatory cytokines was observed to a greater extent over time [15]. Cytokines that were present guided cell polarization [34]. Although one article reported a decrease in pro-inflammatory cytokines and an increase in anti-inflammatory cytokines [22], another reported a decrease in all the cytokines [15]. In the same way, the role of IL-6 remains quite complex. In type 2 diabetes, this cytokine exerts a protective effect by limiting inflammation, and in SS studies, it appeared to have a favorable role. Its role was not beneficial in other pathological conditions [25]. OE-MSCs injected in mice inhibited disease progression. Thus, OE-MSCs induced IL-6 production by myeloid suppressor cells (MDSCs) via toll-like receptor 4 (TLR4) signaling. IL-6 is thus involved in improving the immunosuppressive function of MDSCs with an increase in the production of NO, ROS, and arginase [25]. Another study focusing on MDSCs showed an improvement in MDSC immunosuppressive capacities with increased arginase and NO after their injection in mouse models. The immunosuppressive effect of MDSCs was primarily mediated by TGF-β released by BM-MSCs [26]. Qi et al. revealed two other molecules responsible for the inhibition of the differentiation of MDSCs by UC-MSCs and established the presence of two subpopulations of MDSCs: polymorphonuclear MDSCs (PMN-MDSCs) and monocytic MDSCs (M-MDSCs) (Figure 3). The first was inhibited via PGE2/COX2 and the second via IFN-β. UC-MSCs inhibited the differentiation of MDSCs and increased their immunosuppressive capacities [23].

#### 3.2.3. Role of IL-12

One study looked more specifically at the role of IL-12, a pro-inflammatory cytokine produced by dendritic cells and involved in several autoimmune diseases. This interleukin was found in greater quantities in patients with SS and in the mouse model of the disease. It played a role in the differentiation of naive CD4^+^ T cells into Th1 T cells [34]. Treatment with IL-12 in mouse models showed a resurgence of disease symptoms with increased lymphocyte infiltration into the salivary glands and decreased salivary flow. Injection of MSCs in 10 patients with SS resulted in a decreased production of IL-12 in dendritic cells 7 days after intravenous injection [27]. Two studies demonstrated the correlation between IL-12 and SS [26,35]. In contrast, another study previously observed a decrease in IL-12 in SS patients [36]. The process of differentiation and maturation of DCs would be altered by MSCs, hence the inhibition of IL-12. However, its regulatory mechanism remains largely unknown.

Studies have shown the efficacy of anti-IL-12 antibodies in other autoimmune diseases [37], but before making a hypothesis on the therapeutic role of anti-IL-12 antibodies in SS, further studies are needed.

#### 3.2.4. Other Cytokines

Yao et al. investigated the IFN-β/IL-27 signaling axis. Following MSC injection in patients with SS, they observed an increase in IL-27 produced by dendritic cells and thus a restoration of the Th17/Treg cell ratio. After measuring the levels of CD40, CD137, IFN-α/β, and IFN-γ in the MSCs cocultured with dendritic cells, they demonstrated the importance of IFN-β in the production of IL-27 [29]. Another signaling axis was reported as essential for playing a role in the development of the disease: the SDF-1/CXCR4 axis. A decrease in the transcription of CXCR4 (receptor for CXCL12/SDF-1) was indeed reported in SS patients compared to healthy individuals. SDF-1/CXCR4 is essential for directing BMMSCs in migration to inflammatory sites to control autoimmunity [28].

#### 3.2.5. Impact on Apoptotic Cells

DPSCs led to a decrease in apoptotic cells in the sub-mandibular glands [21]. When comparing DPSCs and BMMSCs, the anti-apoptotic capacities of DPSCs were greater than those of BMMSCs [21].

Murine embryonic MSCs (ME-MSCs) could promote epithelial cell proliferation while suppressing epithelial cell apoptosis in mouse salivary glands [16]. Likewise, MSC treatment decreased the level of caspase3, an apoptotic protein involved in cellular apoptosis [14].

#### 3.2.6. Impact on Treg, Th1, Th2, Th17, and Tfh Cell Responses

DPSCs participated in the induction of FoxP3 expression, a Treg cell marker, and the inhibition of RAR-related orphan receptor (ROR) gamma and GATA binding protein 3 (GATA3) expression, the markers of Th17 and Th2 cells. The study confirmed that DPSCs induced Treg cells and suppressed Th1 and Th17 cells [21,22]. According to Matsumura-Kawashima et al., DPSCs induced T cell differentiation into Treg cells via the TGF-β1/Smad signaling pathway. MSC treatment influenced immunomodulation by inducing more Treg cells [14,30]. Finally, MSC treatment and combination therapy (MSC + complete Freund’s adjuvant CFA) decreased the number of T and B cells in the tissues while increasing Treg cells [19].

Treatment with BMMSCs suppressed the inflammatory response by promoting Treg and Th2 cells while suppressing the Th17 and Tfh responses. Contrarily, by blocking CXCR4 these effects were abolished; thus, the SDF-1/CXCR4 axis played an important role in the immunomodulatory activity of BMMSCs [28]. The differentiation of Tfh cells was inhibited by the secretion of IDO by UC-MSCs in a coculture of UC-MSCs with naive CD4^+^ T cells under Tfh cell-polarizing conditions. Tfh cells played a crucial role in disease development by promoting the differentiation and maturation of B cells. Tfh cells permitted the production of anti-SSB/La antibodies [32]. MSCs derived from Wharton’s jelly in the umbilical cord blood (Cps-hUCMS) suppressed T cell proliferation in SS patients and restored the Treg/Th17 cell ratio that might positively impact SS [31]. After MSC transplantation, there was a decrease in the percentage and absolute number of Th17 and Tfh cells and an increase in Treg cells, but no influence on Th1 and Th2 responses were highlighted in this experiment. Anti-IL-12 antibodies caused inhibition of Th1, Th17, and Tfh cells [26]. BMP6 was overexpressed in BMMSCs of patients with SS and induced an increase in Th1 and Th17 cells [13]. The proliferation capacity of CD4, Th1, and CD8 T cells was reduced after treatment with anti-BMP6 antibodies [13].

In mice transplanted with SHEDs, a decrease in Th1/Th2/Th17 cytokines was observed. SHEDs directed T cell to Treg cells and suppressed Th1 and Tfh cells in the spleen [15]. MSC treatments thus have a positive effect on the disease by decreasing the numbers of Th1, Th17, and Tfh cells and increasing the amount of Treg cells.

#### 3.2.7. Impact on B Cell Response

The selective suppression of B cells, as well as the decrease in the production of anti-SSA/Ro60 autoantibodies, was demonstrated following treatment with DPSCs [22] or MSCs [14]. Interestingly, the amount of B-cell-activating factor (BAFF), a cytokine involved in the survival of autoreactive B cells, was lower in the salivary glands in a mouse model of SS treated with MSCs or MSCs extract (MSCsE) compared to untreated mice [14]. Likewise, a decrease in BAFF was observed after treatment with MSCs and a combined treatment (CFA + MSC) in NOD mice [19]. SS is a disease characterized by a lymphocyte infiltration involving T and B cells. Injection of MSCs into NOD mice had a beneficial effect on the disease by lowering the level of B cells. Indeed, B cells are responsible for the orientation of the immune response through the production of autoantibodies and cytokines, but also play a role as antigen-presenting cells [38]. However, few studies document the impact of MSCs on B cells.

#### 3.2.8. Anti-SSA/Ro and Anti-SSB/La Antibodies

The presence of anti-SSA/Ro autoantibodies is one of the major criteria present in the European League against Rheumatism (EULAR) classification for the diagnosis of SS [39].

Xu et al. showed a decrease in anti-SSA/Ro and anti-SSB/La antibodies one month after UCMSC injection in 24 patients. Other studies [14,22,25,30] also demonstrated the decrease in anti-SSA/Ro antibodies following injection of MSCs in mice. The decrease in anti-SSA/Ro autoantibodies was greater after injection of OE-MSCs exosomes (OE-MSCs-Exos) compared to treatment with BM-MSCs exosomes [25].

### 3.3. Impact on Salivary Flow

Several studies have investigated the impact of MSCs on salivary flow rate. By comparing the salivary flow before and after injection of MSCs in the same mouse, they noticed an improvement after MSC administration [14,15,18,19,22,25,26,27,29].

This improvement was demonstrated from the first week following the injection [29] until 52 weeks after treatment [20].

Four studies showed lower salivary secretion in treated mice compared to the healthy control group. Still, the decrease in salivary secretion is less significant compared to untreated mice [12,16,24,27,28]. These data suggest an improvement in salivary functions after injection of MSCs.

Hu et al. indicated a greater increase in salivary flow after intravenous injection of Id3 knock-out (KO) BMMSCs in mice. Likewise, with anti-BMP6 antibodies, the saliva flow collected at 6 and 10 weeks was greater than that of untreated mice [12]. Conversely, IL-27 KO mice resulted in decreased salivary flow compared to NOD mice. Indeed, IL-27 was found in greater quantities in patients with inactive SS compared to patients with active SS [29]. Unlike IL-27, IL-12 was reported to have a negative impact on salivary flow rate [33]. Transplantation of bone marrow-derived cells also had a great impact on the disease by increasing salivary secretion [40]. Saliva analyzes of alpha amylase activity and protein did not reveal any changes in the composition of saliva [15,19]. However, the electrolyte concentration was modified 44 weeks after injection of BMMSCs into tail vein compared to the concentration at 4 weeks after the injection. Thus, the increase in salivary flow was accompanied by a decrease in Na^+^ and Cl^-^, and a slight increase in K^+^ ions [20].

Xu et al. used the EULAR Sjögren’s syndrome disease activity index (ESSDAI) to demonstrate the activity of SS. There was a significant decrease in this score following the injection of MSCs. The score was calculated at 3, 6, and 12 months and changed from 46% to 71% and finally to 83% of patients with a decreased ESSDAI score greater than 30% [28]. A positive correlation was demonstrated between ESSDAI and IL-12 [26].

From a physiological standpoint, all the studies have demonstrated the existence of a therapeutic effect of MSCs in sicca syndrome, with improved salivary functions. However, the targets and underlying mechanisms differ from one study to another.

### 3.4. Follow Up and Risk of Transplant

The duration of follow-up was different between studies, varying from 2 to 52 weeks in mouse models and up to one year in patients. Similarly, the dose, the time of injection, and the origin of the injected cells differed between the studies. Apart from Liu et al. [30], who administered intraperitoneal MSCs, the other studies used intravenous injections. A single protocol would make studies comparable and facilitate more reliable conclusions.

One study investigated the potential effects of MSC injection by looking for adverse effects and risk of transplant mortality in 404 patients who received MSCs for various autoimmune diseases. The mortality rate was 11.1% after 9 years of follow-up, of which 62.2% of deaths were due to a relapse or complications from the disease [41].

The use of MSCs was believed to be safer than hematopoietic stem cells. The transplanted related mortality (TRM) varies from 5% to 12% after autogenic or allogeneic treatment with hematopoietic stem cell transplantation (HSCT), while it is 0.2% with MSCs. The increased risk of mortality is thought to be due to the conditioning of HSCTs [41]. One of the major complications of HSC injection is graft versus host disease (GvHD). MSCs could have a beneficial effect in the treatment of GvHD after HSC transplantation [42].

All the allogeneic studies concerning mouse models used stem cells from mice not affected by the disease. Xenografts were also found in mice, as they were cells from the human umbilical cord blood injected by veins or intraperitoneally. In human studies, this involved injection of allogeneic umbilical cord stem cells. Among the studies analyzed in this article, none refer to a choice of autologous cells. Mesenchymal stem cells from patients with SS were altered compared to MSCs from healthy patients. Particularly, they presented a decrease in immunomodulatory functions [28].

Therefore, it seems preferable to use allogeneic MSCs in the context of research, especially since they exhibited non-immunogenic characteristics and weak or absent expression of the non-major histocompatibility complexes, MHC-I and MHC-II [43]. Consequently, their low immunogenicity limited the risk of an immune reaction in the host.

The use of induced pluripotent stem cells (iPSCs) could mitigate the risk of rejection by allowing autologous transplants from sick patients [44].

## 4. Conclusions

All analyzed studies demonstrated physiological and cellular changes in mice and SS patients after injection of MSCs. Indeed, there was an improvement in salivary flow, a decrease in lymphocyte infiltration, and a decrease in pro-inflammatory cytokines.

Thus, these cells could evolve into a future therapeutic alternative. Other clinical trials are necessary and must explore alternate ways of introducing MSCs (ophthalmic, nasal instillation, intra-canal injection into the salivary glands, ingestion in capsules, etc.). Even so, the underlying mechanisms remain to be fully understood. Additional clinical studies are needed to provide more powerful scientific evidence, such as long-term effects on clinical outcomes and safety, the duration of therapy, the optimal timing, and the dosages for treating SS.

## Figures and Tables

**Figure 1 ijms-22-10474-f001:**
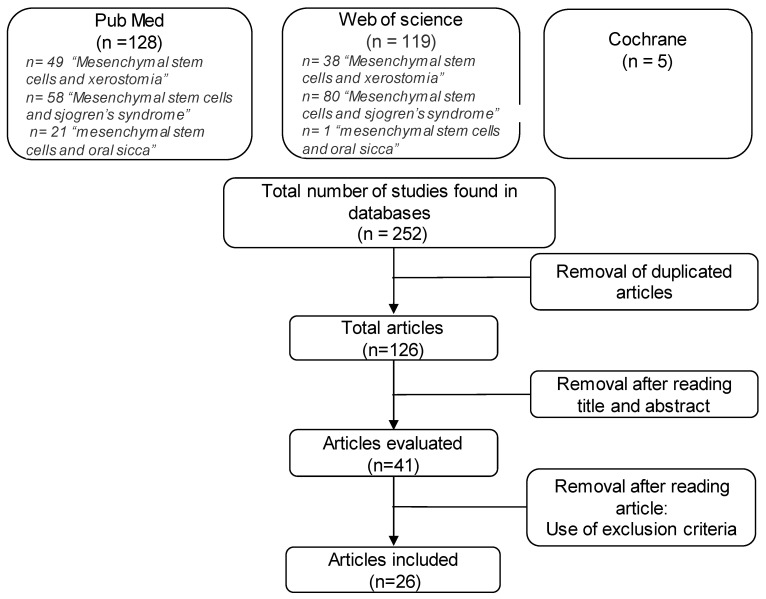
Selection flowchart for articles referring to mesenchymal stem cells in Sjögren’s syndrome.

**Figure 2 ijms-22-10474-f002:**
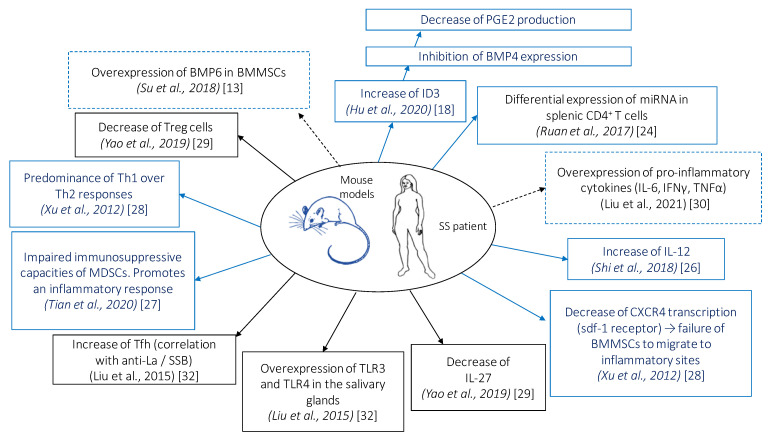
Cellular and molecular modifications induced by mesenchymal stem cell (MSC) administration in patients with Sjögren’s syndrome (SS) and in mouse model of SS. Bone morphogenic protein (BMP), bone marrow MSCs (BMMSCs), DNA binding protein inhibitor (Id), interferon (IFN), interleukin (IL), myeloid suppressor cells (MDSCs), prostaglandin E2 (PGE2), regulatory T cells (Treg), helper T cell (Th), follicular helper T cell (Tfh), toll-like receptor (TLR), tumor necrosis factor (TNF). Solid blue boxes indicate that these observations were made in mice, solid black boxes indicate that these observations were made in humans, and dashed boxes indicate that these observations were made in humans and mice.

**Figure 3 ijms-22-10474-f003:**
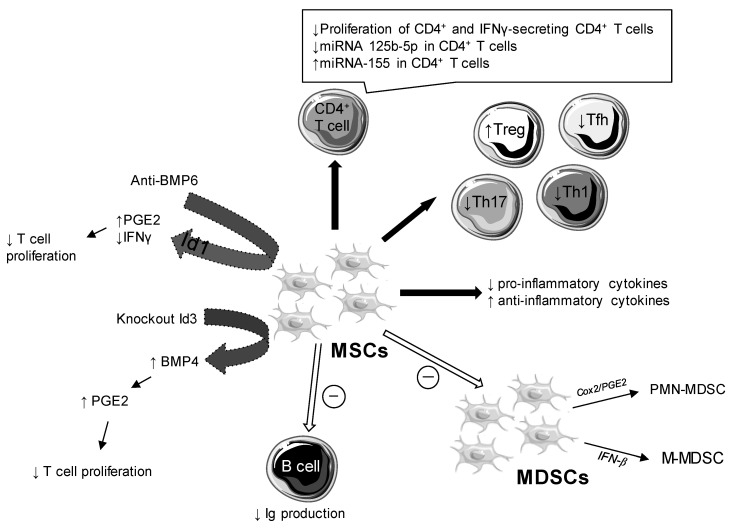
Cellular and molecular impacts of mesenchymal stem cells (MSCs) in Sjögren’s syndrome (SS)**.** Bone morphogenic protein (BMP), immunoglobulin (Ig), interferon (IFN), DNA binding protein inhibitor (Id), myeloid suppressor cells (MDSCs), polymorphonuclear (PMN)-MDSCs, monocytic MDSCs (M-MDSCs), regulatory T cell (Treg), helper T cell (Th), follicular helper T cell (Tfh), prostaglandin E2 (PGE2), cyclooxygenase (Cox).

**Table 1 ijms-22-10474-t001:** Effects of MSC treatment in in vivo mouse models of Sjögren’s syndrome-like symptoms.

Type of MSC	Mouse Model	Administration	Follow Up	Major Effects of MSC Treatment		Ref.
				Increase	Decrease	
OE-MSCs BMMSCs	C57BL/6 mice with experimental SS induced by immunization with SG proteins	2 IV injections (day 18 and 25 after immunization)	Day 42 after immunization	- Salivary flow rate	- Histological score in SG - Anti-M3R and anti-SSA/Ro autoantibody levels in serum (only with BMMSCs) -Production of inflammatory cytokines (IFN-γ and IL-17)	[25]
BMMSCs			Day 35 and Week 15 after immunization		- Histological score in SG - Anti-M3R and anti-nuclear autoantibody levels in serum (only for BMMSCs) - Production of inflammatory cytokines (IFNγ and IL-17)	[27]
DPSCs BMMSCs	MRL/MpJ-faslpr/faslpr (MRL/lpr)	4 IV injections of conditioned-media MSC (twice a week)	Week 2 after last injection	- Salivary flow rate - Production of anti-inflammatory cytokines (TFGβ1, IL-10 and IL-13)	- Focus score of SG - Epithelial cell apoptosis in SG	[22]
DPSCs BMMSCs	NOD mice	4 IV injections of conditioned-media MSC (twice a week)	Week 2 after last injection	- Salivary flow rate - Production of anti-inflammatory cytokines (TFGβ1, IL-10, and IL-13)	- Focus score of SG (only with DPSCs) - Epithelial cell apoptosis in SG	[21]
BMMSCs	NOD/Ltj mice (Cdh23ahl)	IV injections at early stage of SS (week 6 = prevention group) or at developed stage (week 16 = treatment group)	Week 2 after injection	- Salivary flow rate in 2 groups - Production of anti-inflammatory cytokines (TFGβ1, IL-10, and IL-13) in treatment group	- Anti-nucleic, anti-α-fodrine and anti-SSA/Ro autoantibody levels in serum in treatment group - Production of inflammatory cytokines (IFN-γ, IL-6, and IL-17) in treatment group	[28]
Id3-deficient BMMSCs	NOD/ShiLtj mice	IV injection of Id3-deficient BMMSCs or WT BMMSCs	Week 2 after injection	- Salivary flow rate, better with Id3-deficient BMMSCs	/	[18]
ME-MSCs	NOD/Ltj mice	4 IV injections of MSC (twice a week)	Week 2 after last injection	- Salivary flow rate	- Epithelial cell apoptosis in SG	[16]
BMMSCs Extract	NOD mice	4 IV injections (once a week)	Baseline Week 4, 8, 12, and 16 after last injection	- Salivary flow rate - Tear flow rate - Production of anti-inflammatory cytokines (IL-10)	- Anti-SSA/Ro autoantibody levels in serum	[14]
SHED	NOD mice	2 IV injections (once a week)	Baseline Week 1 Week 2	- Stimulated salivary flow rate at week 2	- Focus score of SG at week 2 - Epithelial cell apoptosis in SG	[15]
iPSC-MSCs BMMSCs	NOD/ShiLtj mice	2 IV injections (once a week)	Week 3 after last injection	/	- Anti-SSA/Ro and anti-SSB/La autoantibody levels in serum	[17]
hUCMSCs	NOD mice	IV injection	Week 4	- Salivary flow rate	/	[23]
UCMSCs	NOD/Ltj mice	IV injection	Day 28 after injection	- Salivary flow rate	/	[26]
BMSCs	NOD/Ltj mice	4 IV injections (twice a week)	Week 4	- Salivary flow rate -Production of anti-inflammatory cytokines (TFGβ1 and IL-10)	- Focus score of SG - Production of inflammatory cytokines (IFNγ)	[24]
BMSCs CD45^-^/TER119^-^	NOD mice	4 IV injections (twice a week)	Baseline Week 10 Week 14	- Salivary flow rate	- Lymphocytic infiltrate in SG	[19]
BMSCs	NOD mice	12 IV injections (twice a week)	Baseline Week 2, 12, 34, 38, 44, and 52	- Salivary flow rate (week 12, 34, and 52)	/	[20]
UCMSCs	NOD mice	5 Intraperitoneal injections (once a day)	Week 8 or Week 12	- Salivary flow rate - Production of anti-inflammatory cytokines (IL-10)	- Anti-α-fodrine and anti-SSA/Ro autoantibody levels in serum - Histological score in SG - Production of inflammatory cytokines (IL-6)	[30]
UCMSCs	NOD/Ltj mice	IV injection	Week 4 after injection	- Salivary flow rate	/	[29]

IV: intravenous; SG: salivary gland; SS: Sjögren’s syndrome.

**Table 2 ijms-22-10474-t002:** The in vivo and in vitro studies of MSC treatment in SS patients.

Type of MSC	In Vivo/In Vitro Study	Follow Up	Major Effects of MSC		Ref.
			Increase	Decrease	
UCMSCs	In vivo IV injection (10 patients)	Day 7	- IL-12 levels in serum	/	[26]
UCMSCs	In vivo IV injection (24 patients)	Baseline Week 2 Month 1 Month 3 Month 6 Month 12	- Unstimulated and stimulated salivary flow rate for most of 11 patients	- SSDAI score for all 24 patients and VAS score for most of patients - SSA/Ro autoantibodies for 7 patients - SSB/La autoantibodies for 6 patients	[28]
UCMSCs	In vivo IV injection (38 patients)	Week 1	- IL-27 and TGFβ levels in serum - Tregs among PBMCs	- IL-17-expressing T cells among PBMCs	[29]
	In vitro Coculture with monocyte-derived DCs	Unknown	- IL-27 production of DCs	/	
UCMSCs	In vitro Coculture with PBMC from SS patients or from HC	Day 4	/	- Expansion and differentiation of Tfh from PBMC from SS patients	[32]
UCMSCs free or encapsulated	In vitro Coculture with PBMC from SS patients or from HC	Day 4	- Tregs among PBMCs (with encapsulated UCMSCs)	- Proliferation of T cells from SS patients (only with encapsulated UCMSCs)	[31]
UCMSCs	In vitro Coculture with CD4^+^ T cells from blood of SS patients or of HC	Day 3	- IL-6 production in supernatants - Upregulation of 13 miRNA whose miR-115–5p having impact on TCR signaling pathway	- Proliferation of T cells from SS patients - IFNγ production in supernatants	[33]
BMMSCs	In vitro Coculture of PBMC from HC with BMMSCs from HC or SS patients	Day 4	- T cell proliferation with BMMSCs from SS patients compared to BMMSCs from HC	/	[28]

DCs: dendritic cells; IV: intravenous; HC: healthy control; PBMCs: peripheral blood mononuclear cells; SS: Sjögren’s syndrome; SSDAI: Sjögren’s syndrome disease activity index; TCR: T cell receptor; Tfh: follicular T helper cell; Tregs: regulatory T cells; VAS: visual analogue scale.

## Data Availability

Not applicable.

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
