# Peer review of "Treatment of Sjögren’s Syndrome with Mesenchymal Stem Cells: A Systematic Review"

_ijms, 2021, doi:10.3390/ijms221910474_

Round 1

Reviewer 1 Report

The article needs significant language editing as I have encountered many week, long, and/or poorly constructed sentences.

The abstract is poorly written and contains incorrect information, for example, MSCs were first isolated form fatty tissue! Thorough literature review regarding this topic is strongly recommended.

The introduction is short and poorly constructed; the first few short paragraphs can actually be grouped into one only. 

The data obtained from the analysis is better be organized in a table; it will help the readers know which paper included what.

The title should be more precise such as treatment of SS with MSCs......etc.

Results from human studies should be separated from animal ones.

The authors mentions that there are three other studies that have discussed the same topic, however, they did not indicate how their paper is different from theirs, otherwise it is just a repetition.

Thank you

Author Response

We have made the necessary changes and addressed the issues raised by the reviewers and would like to highlight in a point-by-point manner the changes made and how raised issues and concerns were addressed.

1-The article needs significant language editing as I have encountered many week, long, and/or poorly constructed sentences.

The manuscript has been language edited.

2-The abstract is poorly written and contains incorrect information, for example, MSCs were first isolated form fatty tissue! Thorough literature review regarding this topic is strongly recommended.

The abstract has been revised and corrected.

3-The introduction is short and poorly constructed; the first few short paragraphs can actually be grouped into one only. 

We thank the reviewer for his suggestion and we have grouped the first two paragraphs.

4-The data obtained from the analysis is better be organized in a table; it will help the readers know which paper included what.

We agree with the reviewer comment. Consequently, two additional tables have been added. The first one focuses on the effects of the different kinds of MSCs in in-vivo mouse models. The second one focuses on the effects of the different kinds of MSCs on SS patients.

5-The title should be more precise such as treatment of SS with MSCs......etc.

The title has been modified as suggested by the reviewer.

6-Results from human studies should be separated from animal ones.

We agree, see response to comment 4. This point is also illustrated in Figure 2.

7-The authors mentions that there are three other studies that have discussed the same topic, however, they did not indicate how their paper is different from theirs, otherwise it is just a repetition.

As an expert of the field, the reviewer knows perfectly that our review is not a repetition. Indeed, three other reviews had previously summarized the effectiveness of MSCs in the treatment of sicca syndrome and particularly SS. The latest literature review published in 2020 addressed biological therapies and only presented a small part of the use of MSC in the treatment. The two others (Chen et al., 2018 and Jensen, 2014) are far from containing the latest advances in the field because more than 10 articles have been published since 2018. Consequently, to avoid any confusion, this sentence has been removed.

Reviewer 2 Report

In results sections, for each subheading, an introductory statement will be helpful.

On  "Impact on pro- and anti-inflammatory cytokines", The description from lines 132 to 152 is confusing. Not sure what message the authors wish to convey. Please organize the studies in some order. Seperating in vitro and in vivo studies will be helpful. Same suggestion applies to the next subheading.

For "impact on salivary flow" please provide a table that provides a snapshot of all studies.

Reference #2 is in French, please provide an alternative 

 Authors may wish to consider combining results and discussion to keep the flow of the review.

Author Response

We have made the necessary changes and addressed the issues raised by the reviewers and would like to highlight in a point-by-point manner the changes made and how raised issues and concerns were addressed.

1-In results sections, for each subheading, an introductory statement will be helpful.

We thank the reviewer for his comment. In the revised manuscript an introductory statement has been added.

2-On  "Impact on pro- and anti-inflammatory cytokines", The description from lines 132 to 152 is confusing. Not sure what message the authors wish to convey. Please organize the studies in some order. Separating in vitro and in vivo studies will be helpful. Same suggestion applies to the next subheading.

We thank the reviewer for his comment. Two additional tables have been added. The first one focuses on the effects of the different kinds of MSCs in in-vivo mouse models. The second one focuses on the effects of the different kinds of MSCs on SS patients.

3-For "impact on salivary flow" please provide a table that provides a snapshot of all studies.

This information is now clearly available in the new Table 2.

4-Reference #2 is in French, please provide an alternative 

Reference #2 in, French, has been removed and an alternative has been added.

5- Authors may wish to consider combining results and discussion to keep the flow of the review.

We thank the reviewer for his comment. In the revised manuscript we have combined results and discussion.

Round 2

Reviewer 1 Report

The article has beed edited as directed before, however, there are some issues that need to be addressed before publication. 

  1. the abstract should be organized into one paragraph only.
  2. I have detected some grammar and punctuation mistakes through out the manuscript. I advise the authors to have it professionally edited.
  3. the second short paragraph in the introduction should be added to the first one.

Author Response

  • the abstract should be organized into one paragraph only.

We have modified the abstract accordingly.

2- I have detected some grammar and punctuation mistakes through out the manuscript. I advise the authors to have it professionally edited.

The manuscript has been professionally edited. The professional has looked at academic tone and language; paragraph and sentence structure; correctness and consistency of elements such as spelling, grammar, capitalization, and punctuation; reducing repetition; and overall coherency.

Though the use of commas is often subjective, US English and academic writing typically call for using serial commas. In this paper, since serial commas can be helpful for improving clarity, especially in US English, and to maintain consistency, the manuscript has been revised to use serial commas throughout.

  • the second short paragraph in the introduction should be added to the first one

We have modified the introduction accordingly.